# Gender Gap in Self-Rated Health: A Cohort Perspective in Eastern European Countries

**DOI:** 10.3390/healthcare10020365

**Published:** 2022-02-13

**Authors:** Marta Gil-Lacruz, Ana Isabel Gil-Lacruz, Jorge Navarro-López, Isabel Aguilar-Palacio

**Affiliations:** 1Bienestar y Capital Social (BYCS), Department of Psychology and Sociology, Health Science Faculty, University of Zaragoza, 50009 Zaragoza, Spain; 2Bienestar y Capital Social (BYCS), Department of Management, School of Engineering and Architecture, University of Zaragoza, 50018 Zaragoza, Spain; anagil@unizar.es; 3Grupo Decisión Multicriterio Zaragoza (GDMZ), Department of Applied Economics, Faculty of Economics and Business, University of Zaragoza, 50005 Zaragoza, Spain; jnavarrol@unizar.es; 4Grupo de Investigación en Servicios Sanitarios de Aragón (GRISSA), IIS Aragón, Department of Microbiology, Preventive Medicine and Public Health Medicine Faculty, University of Zaragoza, 50009 Zaragoza, Spain; iaguilar@unizar.es

**Keywords:** health gender gap, self-rated health, wealth, economic inequity, health international comparison, European Health Interview Survey, health cohorts research

## Abstract

Background: The relationship between self-rated health and gender differs across countries and generations. The aim of this study is to analyze the effect of socioeconomic conditions on self-rated health from a generational perspective, its differential effect on gender, and its influence on the gender gap in order to explore health diversity using a multidisciplinary approach and considering policy implications in Eastern European countries. Methods: We used data drawn from the European Health Interview Survey for eight Eastern European countries and EUROSTAT from 2006 through to 2009. We conducted multilevel analyses to understand the individual and national health determinants of self-rated health by gender and to determine whether national differences remain after controlling for micro variables. In order to analyze the role of equity (Gini quartile) in gender differences, Oaxaca analyses were used. Results: The self-rated health gender gap increases with age. Individual characteristics, such as educational level or smoking, influence citizens’ perceived health, and have a stronger effect on women than on men. Knowing both the characteristics (endowment effects) and the effects of individual characteristics (coefficient effects) on health is important in order to understand gender gaps among people from the silent generation. Conclusions: Our research indicates that random effects are greater for men than for women. Moreover, random effects might be explained to a certain extent by economic equity (Gini index). The combined effects of gender, cohort, and geographical differences on self-rated health have to be taken into account to develop public health policies.

## 1. Introduction

Self-assessed health is an important indicator of people´s health status. It is a comprehensive assessment of an individual’s physical, mental, and social well-being [1]. A negative self-perception of health is related to worse indicators of acute and chronic morbidity [2], a greater degree of disability, more frequent use of health resources [3], and deficiencies and/or difficulties in the accomplishment of daily work, domestic, and recreational activities [4]. Despite the subjectivity of this measure, the behaviors to which it leads are observable in clinical practice. In this way, self-reported health is a good starting point to study citizens’ health [5,6].

Social characteristics, such as socioeconomic status or educational level, influence the self-interpretation of health [7,8,9,10,11]. The influence of these variables tends to occur in an interdependent way [12]. Socioeconomic status can influence health outcomes—for example, through access to medical care—and in this way can be used to predict self-assessed health [7]. Research on income inequality provides empirical evidence about the existence of a social gradient in health [13]. On average, people with low socioeconomic status have worse health than people in a position of wealth. In this way, unequal societies tend to have unequal health [14].

Additionally, it has been argued that disadvantaged groups will fail to perceive and report the presence of illness or health deficits, which may result in misleading assessments of population health, which is called “reporting heterogeneity”. On the one hand, advantaged populations tend to report higher levels of poor health than disadvantaged populations [15,16]. On the other, self-reported questions may have led to the underestimation of unhealthy lifestyles such as smoking, overweight, and low physical activity, because people tend to report their lifestyles to be better than they are [17].

Socioeconomic macro conditions also have an important influence on self-assessed health. In this sense, the scientific literature has focused on the influence of the inequality index of the distribution of resources (Gini coefficient) [18]. It seems clear that in rich and equitable countries, policy makers have more resources to spend on health-promoting social programs and their citizens have higher incomes to invest in health [19]. In the European context, data from the European Social Survey (2002, 2004, and 2006) and Huijts, Monden, and Kraaykamp (2010) [20] show that countries’ GDP per capita is positively associated with individuals’ self-rated health. However, this depends on the starting point of domestic wealth. In this sense, Deaton (2003) [21] studied how an additional dollar in public health investment has a stronger effect in poor countries than in richer ones. One reason for this could be that economic growth allows more people to escape poverty in economically underprivileged countries [22]. In fact, income inequality could be a key variable that helps to explain these results. A literature review by Wilkinson and Pickett (2006) [23] showed that over 70% of 168 studies reported worse health in countries with large income inequalities. In spite of this trend, this relationship is not as evident in the European context as it is in less egalitarian countries such as the United States [24].

Cultural beliefs, attitudes, and values also play an important role in the self-perception of health [25]. In this sense, gender is a relevant variable associated with culture. For example, studies show that Latin culture is more permissive towards women expressing a negative perception of health, but these results may not have a unidirectional interpretation [4,26]. International research carried out across 17 European countries found that although this perception changes widely among nations, women consistently have poorer self-reported health than men, even in healthy countries [27], and the gender gap is wider in eastern and southern EU countries [28]. Gender inequalities in health have been explained as the result of a stratification process that assigns different socioeconomic opportunities to men and women [29]. Educational attainment benefits women more than men in terms of self-rated health [30]. The major presence of women in the labor market also contributes to better female self-evaluation [31]. In fact, the potential positive effects of education on women’s health can be reduced when the gender gap decreases [32]. In order to promote better self-rated health, research must focus on the different impact of socioeconomic variables by gender.

The relationship between gender differences and culture in self-rated health is not static. In this sense, it is important to differentiate between the effects of age and cohort [33]. On the one hand, age effects are related to the individual aging process. Chronological age determines our health as the years go by, regardless of our generational cohort. On the other, the effects of birth year, and consequently, the effects of sharing life events and socioeconomic context for a group are measured by cohort effects [34]. Generational cohorts help us to understand why people from different generational cohorts value their health differently when they reach the same chronological age. For example, the research conducted by Etherington (2017), which used longitudinal data from the Panel Study of Income Dynamics, found that prior to analyzing cohorts’ effects on self-rated health, women reported worse health than men did. However, when introducing cohort variables, no gender difference was found, with the exception of the oldest members of the population (born from 1924 to 1933) [35]. Therefore, differences not only between populations but also between cohorts may exist [36,37,38].

Regarding the dynamic variation in the educational gradient in health, Delaruelle et al. introduced two theoretical hypotheses: a diminishing health returns model and a cohort accretion hypothesis. The first predicts a decrease in educational disparities in health, while the second one affirms that the education–health gap will be bigger in younger cohorts [39]. The trend of poor self-reported health in younger birth cohorts is deeply worrying for the individuals affected, and may also have a negative impact on the health care system. For example, in spite of the increased levels of education, higher incomes, and lower smoking rates among baby boomers in relation to people from the silent generation, these improvements may be counterbalanced by the adverse effect of increasing body mass [40].

Therefore, further research on the direct effects of gender and culture on self-assessed health is needed [41]. In this paper, we attempt to contextualize this need in the research on the gender gap in self-rated health in Eastern European countries.

We focused on Eastern European countries for several reasons. In spite of the significant geographical size and population of Eastern European countries, we do not have enough knowledge about the evolution of inequities in regard to self-reported health and gender in these countries [42]. Eastern and Southern European countries have a larger gender gap than the rest of the EU countries [28]. The common sociological, political, and cultural elements of these countries due to their similar histories during the 20th century could be the root of the traditional and conservative view of the role of women in these regions, which identifies them as the person who should be in charge of caring for the family. In the last few decades, transitional pluralist state structures, the prevalence of traditional values, and the lack of women playing active roles in politics have contributed to the need for more research about health in these countries [43]. Sex and cohort differences could influence the experience of health and wellbeing. International comparisons in this big region are fundamental in order to address health disparities, as they are useful for understanding the impact of policies on the social determinants of health [44]. Eastern European countries represent a group of welfare states, with the highest number of countries according to the European Health Interview Survey. All of these eight countries share many similarities. Looking at countries characterized by the same welfare regime allows us to identify, isolate, and study differences in the micro and macro data.

Thus, the main objective of this study is to analyze the effect of socioeconomic conditions on self-rated health from a generational perspective, its differential effect on gender, and its influence on the gender gap in Eastern European countries.

## 2. Materials and Methods

The data were drawn from the first wave of the European Health Interview Survey (EHIS 83/2014). The selection of countries that share important characteristics (welfare state and capacity to create richness) allowed us to identify how differences (equity) lead to different health statuses. The use of standard questionnaires and guidelines in all countries guaranteed the comparability of our findings [45].

The final sample size was 79,487 individuals living in eight Eastern European countries (Bulgaria, the Czech Republic, Hungary, Latvia, Poland, Romania, Slovenia, and Slovakia). The sampling units used in Slovenia, the Czech Republic, Hungary, and Latvia were individuals. In Bulgaria, the sampling units were households, and in Romania, Poland, and Slovakia the sampling units were dwellings. Estonia was not included due to the lack of information about some of the key variables examined in this research.

In this study, we considered people from the age of 15 years old. The variable age refers to the chronological age in the survey year, while the generational cohort was calculated according to birth year intervals. We repeated estimations by gender and generational cohort. Although there are slight international differences regarding the birth year of generation cohorts, we considered the following aggregation [46]: the silent generation (people born between 1924 and 1945), baby boomers (1946–1964), the X generation (1965–1980), and the Y generation (1981–1990). This information was obtained taking into consideration the survey year and the imputed age of each individual.

We selected primary questions related to whether people reported a low state of health (LowSelfRatedHealth). The original question was: How is your health in general? The answers were: it is very good (1), good (2), fair (3), bad (4), very bad (5), do not know (8), and refuse to answer (9). We recodified this variable into LowSelfRatedHealth so that answers were assigned a value of 1 if individuals reported answers from 3 to 5 and otherwise they were assigned a value of 0. Answers 8 and 9 were considered missing values.

As an empirical strategy, we considered multilevel models (STATA: xtmelogit). Multilevel regression models are used when there is a hierarchical structure in the levels of data, with a single dependent variable measured at the lowest level and a set of explanatory variables on each of the levels. We followed the empirical strategy used by Pinilla et al. (2002) [47].

In our case, our data were structured with j-countries, in each of which nj persons were interviewed. Our dependent variable was LowSelfRatedHealth ij, which summarizes if individual i of country j reports a low state of health (1: yes; 0: otherwise). Thus, we represented this variable as:〖LowSelfRatedHealth〗_ij = + u_ij + ε_ij(1)
where X, a set of explanatory variables, includes K regressors. As explanatory variables, we considered socio-demographic variables (age, civil state, educational level, and working status) and lifestyles (smoking, overweight, and sedentarism). The parameter represents the fixed effects, which are dependent on L national variables, K-1 individual variables, and a constant. This model assumes that the random effects uj are distributed normally with a mean of 0 and variance, which stands for differences in the variable LowSelfRatedHealth attributable to the country. It also assumes that the error component εij is also distributed normally with a mean of 0 and variance. Finally, it assumes that the random effects uj and the error component εij are independent, and that the εij are all independent of one another.

As explanatory variables, we controlled for socio-demographic characteristics (age, marital status, educational level, and employment) and lifestyles (smoking, overweight, and lack of physical exercise). Individual socioeconomic characteristics and lifestyles are traditionally included in a literature review to explain self-rated health with individual data. Nevertheless, their inclusion provides further empirical evidence about whether country differences are based on differences in individual, national, and contextual variables. In addition, the information provided by individual and national economic variables completes the whole picture. For example, self-perceived health depends on the family income level as well as on the national equity level and welfare regime.

## 3. Results

### 3.1. Descriptive Results

Table 1 summarizes the main descriptive statistics.

Older individuals reported a worse self-perceived state of health than younger ones, and women reported worse states of health than men. The gender gap increased with age. One potential reason for this is that life expectancy is higher for women; thus, living longer does not mean living better. In addition, another consequence is that the percentage of men who are married in the silent generation is much greater than the percentage of their female counterparts, among whom the percentage of widows is greater. Education levels improved among younger generation cohorts. We had to be careful with the case of the Y generation, because most of them were still enrolled in the education system (over 40%). For the silent generation and baby boomers, the education level was higher for men than for women, whereas this situation was reversed among younger generation cohorts. There were also important gender differences for working situation across the generations. The employment rate was greater for men than for women, and the other way around for the case of homemakers. One consequence of this is that the percentage of pensioners was greater among men than among women of the silent generation. Regarding lifestyles, men were more likely to be daily smokers than women, and the prevalence peak was reached in the X generation. Men were more likely to be overweight than women, and the prevalence peak was reached for baby boomers among men and for the silent generation among women. Lastly, women were found to be more sedentary than men. Gender gaps were flattened out for older cohorts.

As expected, women considered their state of health to be worse than men for all generation cohorts. Self-rated health becomes worse with age, with remarkable gaps between men and women from the silent generation in comparison to baby boomers and for baby boomers in comparison to men and women from the X generation (see Figure 1).

We drew macro data from EUROSTAT. The Gini coefficient was used to measure inequality. The results are provided in Table 2. The Czech Republic, Slovakia, and Slovenia are characterized by high levels of inequality (and are also the most unequal inside the European Union), while Latvia and Romania are the most equal (with Spain and Greece being among the most equal countries inside the European Union). To distribute countries according to the Gini index, we took into account Gini tertiles; thus, countries included in the first group were the most unequal and countries included in the third groups were the most equal.

Looking at the national inequity levels, we observed that people living in countries in the second tertile of inequity showed the best perceived health, while people living in countries with the highest inequity or in the third tertile showed a worse self-perceived state of health (see Figure 2).

### 3.2. Empirical Strategy

As our dependent variable (LowSelfRatedHealth) is binary, we applied multilevel analysis with a logistic function. The multilevel analysis helped us to understand individual and national health determinants and whether national differences remained after controlling for micro variables. We repeated estimations using sub-samples of men and women. We also repeated estimation by generation cohorts.

We were also interested in whether equity (Gini tertiles) plays an important role in health in general and in the corresponding gender gap in particular. We chose to focus on Gini and not on GDP pc because GDP pc is usually an alternative way of measuring average family income, so we prioritized equality versus a second measure of wealth. When classifying countries by Gini tertiles, the Gini values of the two countries included in T2 were close to the highest value of T1 (for the case of Hungary) and the lowest value of T3 (for the case of Bulgaria). Nevertheless, the tertiles, in addition to classifying countries by groups, order groups by values; thus, the comparisons between T1 and T3 remain valid. To that end, we carried out an Oaxaca analysis for LowSelfRatedHealth to address gender gaps, repeating estimations by sub-samples of countries distributed in Gini tertiles. The Blinder–Oaxaca decomposition showed that the differences in the means of LowSelfRatedHealth between men and women were due to differences in the mean values of the independent variables within them and/or group differences in the effects of the independent variables.

Table 3 summarizes the main results for individual characteristics and random effects. Age is an important predictor of the self-perceived state of health, even among people who belong to the same generation. Therefore, older men from the silent generation perceived their health to be worse than younger men from the same generation. Male baby boomers showed a worse health self-perception with age than their female counterparts, whereas age played a greater role for women from the silent generation than for their male counterparts. Civil status had a different impact by gender and generation cohort. As an illustration, being a widow had a stronger impact on women from the silent generation than for men, whereas the impact was stronger on men from the X and Y generations than for women. The impact of being a widow was stronger for younger generation cohorts than for older generations. 

Educational level has an important impact on the self-perceived state of health; thus, higher education levels improved the level of self-perceived health. The impact of education was the greatest for baby-boomers. In general, health perception was more sensitive to education among women than among men. Regarding employment situation, being unemployed versus being employed exerted a negative influence on the self-perceived state of health, with the impact being stronger for men and older generations. Being a pensioner and housekeeper also exerted a negative influence on the self-perceived state of health. Only being a student versus working had a positive influence for men from the Y generation.

Smokers from younger generations perceive their health as worse than non-smokers. A special case is male smokers from the silent generation, who consider their health to be better than non-daily smokers. Being overweight and physically inactive is also negative for health. The negative impact of smoking and overweight on health perception is stronger for women than for men and for younger generations than for older generations. Nevertheless, the negative impact of sedentarism on health perception is, in general, stronger for men than for women from the oldest and the youngest generations.

Random effects are relevant and statistically significant for both male and female sub-samples. Random effect models assist in controlling for unobserved heterogeneity; thus, unobserved heterogeneity plays a greater role in understanding self-perceived health. Random effects are stronger for older generations among women and for younger generations among men. As shown in Table 4, we tried to find an explanation for the random effects based on national differences according to the distribution of Gini tertiles.

Table 4 ratifies the gender differences shown in Figure 2. People living in countries in the second tertile of inequity showed the best perceived health for all generations except the silent generation, for whom a higher level of inequity is associated with worse health self-perception. People living in countries with the highest inequity showed worse self-perceived states of health. Gender differences increased in the older generation cohorts. The widest gender gaps were for people living in the most equal countries among the silent generation and for people living in the most unequal generation among the baby boomers. For people from the X and Y generations, the widest gender gaps were found for people living in countries in the 2nd tertile of the Gini index. Gender differences among the silent generation can be explained by endowment and the coefficient effects for the sub-samples of Gini T1 and T2; however, for people living in most unequal countries, gender differences are not so much explained by differences in the characteristics of men and women, but by differences in the impact of these characteristics. This means, for example, that the educational levels of men and women from the silent generation in the most unequal countries are not so different, but that there is an impact of education on their self-perceived health. In general, the endowment effect plays a negative role in increasing the gender gap in favor of men in the older generation cohorts, but it seems that female characteristics improve over the generations; thus, the endowment effect plays an interesting role in gender gaps in favor of women. The coefficient indicates the negative role of increasing gender gaps in favor of men for all generations, but especially for the silent generation.

## 4. Discussion

Among the results obtained for a sample of Eastern European countries, we want to highlight that age is an important predictor of self-rated health, even among people who belong to the same generation. As in previous research, educational level is positively correlated with the self-perceived state of health, whereas unhealthy lifestyles and poor working conditions are negatively correlated. This research stresses that the impact of education on health perception is stronger for women than for men. Among generation cohorts, education has the greatest impact on baby boomers, as other research has pointed out [40].

The impact of smoking and overweight is stronger for women than for men and for younger generations than for older generations. However, the opposite is true in the case of sedentarism. The European Commission (2013) [48] reports that these findings reflect the influence that stressful life conditions have on both health and health-related behaviors among vulnerable population groups. In addition to work (first shift) and housework (second shift), women have to deal with other issues, such as solidarity, calendar management, diversity and equity work, and the wellbeing of family and friends (third shift) [49]. Each one of these tasks can generate gender issues, but together they contribute to generating stress and discomfort [50,51]. Women’s lives and duties can vary across generations and cultures. The differences found in youth are likely to be maintained in adulthood, including the persistence of sexist attitudes regarding the expression of discomfort, differential treatment of ailments, and work and domestic work [12]. In some way, these results support the particular effects of age and generational cohorts on self-rated health and the gender gap as a factor mediating the impact of educational level on self-rated health [32,39]. The educational and health system should take the corresponding preventive measures based on a model that promotes gender equality throughout the life cycle.

One of the main implications of this research, is the importance of to taking into account differences based on gender, generational cohorts, and national settings. It is significant that people who poorly assess their health are people living in the most unequal and the most equal countries. Regarding healthy aging, people from the silent generation show the widest gender gaps in relation to self-perceived health. In this case, characteristics and their impact on health explain the gender differences in equal countries, but in the case of the most unequal countries, gender differences based on characteristics (endowment effects) are not as important as gender differences based on characteristics’ impact (coefficient effects) [48]. Gender equality and income equality usually go hand in hand when defining self-rated health, but in this research, we only confirmed this result for baby boomers. Precisely the opposite occurs for the silent generation. However, the silent generation is also the cohort that presents the greatest gender differences in relevant variables such as educational level (endowment effect), and in fact these variables also present the greatest gender differences in regard to their effects (coefficient effects).

The scientific analysis of these health inequalities at the national level is a complex topic because we need to isolate the effect of several indicators that are interrelated. In this paper, we selected Eastern European countries because they share a common welfare system, and there is an important linkage between equity and the capacity to create richness. However, in these Eastern economies, health care resources are limited and universal access, a feature of communism, has been lost [52]. The transition to market economies has evolved in parallel with the deterioration in public health resources [53]. The adult population reports poorer self-rated health compared to other European Countries [54]. The fact that the gender gap is greater in the elderly population of the Eastern European countries [55] and lower in other Bismarckian countries could be explained by the protective effect of the pension system, the provision of social resources, and long-term planning [56].

In spite of the results, inequity could be studied as a mediating variable that also affects structural and cultural changes [14]. How income inequality harms population health is still a matter of open debate [24]. In fact, this analysis relates to broader debates concerning progress and sustainability [14,57].

Although self-reported health may not always be able to accurately capture variations in absolute health across countries, it can be especially useful for comparing national states of health among countries [58]. In addition, the ease, speed, and economy of collecting self-reports of health (even with a single-item global question such as the one used here) make such collection attractive for rapid appraisals. Additionally, collecting self-reports of health will make it easier to empirically assess the epidemiologic associations between various exposures and health, especially in countries where objective health data are lacking [58].

For future research, the limitations of the use of self-perceived health assessment might be handled by including objective health indicators such as limitations in daily activities. Another research direction might be to analyze the impact of domestic health systems on the perception of health. In this sense, differences in coverage and access may be relevant in the assessment of citizens’ health. A longitudinal approach to these questions—for example, by a panel data strategy based on individuals, regions, and nations—would provide information on the impact of the economic crisis and its possible recovery according to wealth indicators.

Lastly, a critical point when implementing the multilevel technique is related to the number of observations made at different levels. In this study, we compiled 79,487 observations of individuals living in eight countries. For studies such as this one, it is recommended to include around 20 countries. Nevertheless, the optimal number of groups also depends on the target of inference, because using a large sample of individuals (level 1) in a reduced number of countries (level 2) provides robust estimations of fixed effects, whereas estimated random effects might be biased [59]. Under these circumstances, random effects models might be able to provide better estimations of coefficients at level 1 than models controlling only for fixed effects [60]. Although random effects introduce bias, which implies an overestimation of random effects coefficients, we basically use random effects estimations to identify if controlling for random effects might be important and to compare if there might be a gender gap. The magnitude of the estimated coefficients has not been the object of attention. Potential researchers should discuss the results and how they can be interpreted from the perspective of previous studies and the working hypotheses. These findings and their implications should be discussed in the broadest context possible. Future research directions may also be highlighted.

## 5. Conclusions

Despite the important role of unobserved heterogeneity in determining self-perceived health, especially for men and older generations, our results show that in Eastern Europe, a gender gap in self-assessed health persists with relevant differences in health determinants among different generational cohorts, even when controlling for countries with the same welfare system. Historical, sociocultural, political, and economic context is therefore critical in attempts to understand the health trajectories of women and men, which are not uniform across cohorts [35]. Family income level is an important predictor of self-rated health, which is boosted by the national promotion of income equality. Wealth is important but, by itself, does not explain citizens’ well-being, and other concepts such as income equity must be controlled in order to obtain a better panoramic view of the population under study. In order to narrow the self-rated health gender gap, policy strategies must take into consideration generational and age cohorts and the impact of different living conditions, social representations, attitudes, and values on perceptions of health.

## Figures and Tables

**Figure 1 healthcare-10-00365-f001:**
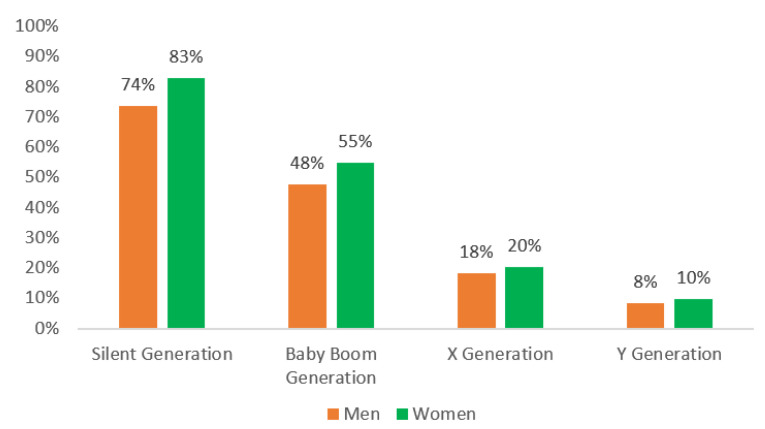
Prevalence of low self-rated health among men and women. Data are given in percentages. Source: self-elaborated graph with data drawn from the European Health Interview Survey and Eurostat.

**Figure 2 healthcare-10-00365-f002:**
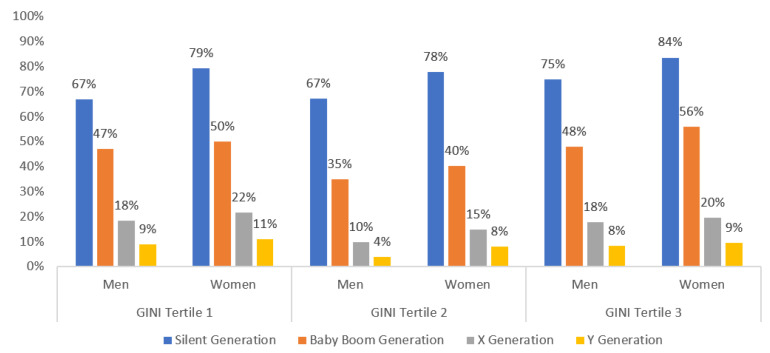
Prevalence of low self-rated health among men and women of different generations by Gini tertiles. Data are given in percentages. Source: self-elaborated graph with data drawn from the European Health Interview Survey and Eurostat.

**Table 1 healthcare-10-00365-t001:** Descriptive statistics by generation stratified by sex (source: European Health Interview Survey).

Variable	Sample by Generation Cohort and Gender							
	Silent Generation (1921–1946)		Baby Boom (1945–1964)		X Generation (1965–1980)		Y Generation (1981–1990)	
	Men	Women	Men	Women	Men	Women	Men	Women
Low self-rated health	74%	83%	48%	55%	18%	20%	8%	10%
Age (mean)	73.1	74.0	54.2	54.5	36.9	36.8	21.8	21.8
Married	74%	36%	80%	70%	70%	74%	13%	23%
Single	3%	3%	9%	5%	25%	15%	86%	75%
Widow	20%	57%	3%	15%	1%	2%	0%	0%
Divorced/separated	4%	4%	8%	10%	5%	9%	0%	1%
Low Education	30%	45%	10%	13%	6%	5%	8%	8%
Intermediate	59%	49%	78%	73%	78%	71%	81%	76%
High Education	11%	7%	12%	14%	17%	24%	11%	17%
Employed	4%	3%	63%	46%	89%	76%	46%	36%
Unemployed	0%	0%	7%	5%	7%	8%	9%	8%
Student	0%	0%	0%	0%	0%	0%	43%	46%
Pensioner	95%	92%	29%	39%	3%	2%	1%	1%
Homemaker	0%	3%	0%	8%	0%	13%	0%	8%
Other	0%	2%	1%	2%	0%	1%	1%	1%
Daily smoker	14%	4%	36%	19%	37%	20%	24%	13%
Overweight	60%	57%	64%	57%	55%	33%	26%	12%
Sedentarism	35%	40%	20%	26%	13%	20%	18%	28%

Number of observations by birth cohorts: silent generation: 16,662 (6640 men and 10,022 women); baby boomer: 26,680 (12,373 men and 14,307 women); X generation: 18,596 (8981 men and 9615 women); Y generation: 17,549 (8887 men and 8662 women).

**Table 2 healthcare-10-00365-t002:** Macro-data (source: EUROSTAT).

Country	Survey Year	Gini ^1^	Gini ^2^	GDP ^3^	GDP ^2^
Bulgaria	2008	30.35	2	4.9	1
Czech Republic	2008	25.15	1	14.1	3
Hungary	2009	25.95	2	9.3	2
Latvia	2008	37.7	3	11.2	2
Poland	2009	30.9	3	8.2	1
Romania	2008	34.9	3	6.9	1
Slovenia	2007	23.43	1	11.8	3
Slovakia	2009	25.45	1	17.4	3

^1^ Gini index for the completion year of the survey. If multiple measurements were available, or there were different sources of information for the same year, the average was taken. ^2^ Distribution by tertiles. ^3^ Gross domestic products at market prices. Unit: current prices, euro per capita (×1000).

**Table 3 healthcare-10-00365-t003:** Low self-rated health multilevel by country.

	Silent Generation		Baby Boomers		X Generation		Y Generation	
	Men	Women	Men	Women	Men	Women	Men	Women
**Fixed Effects**								
Age	0.555 ***	0.581 ***	0.615 ***	0.368 ***	−0.418 **	0.320 **	−0.141	0.219
Age^2^	−0.003 ***	−0.004 ***	−0.005 ***	−0.003 ***	0.007 ***	−0.003	0.004	−0.004
Married ^a^	--	--	--	--	--	--	--	--
Single	−0.257	0.015	0.151 **	−0.059	−0.032	0.027	0.068	0.196
Widow	−0.058	0.105 *	−0.079	0.076	0.850 **	0.362 **	1.552 **	−0.071
Divorced/separated	−0.052	−0.144	0.063	−0.016	0.037	0.160 *	−0.585	0.980
Low Education	0.934 ***	1.026 ***	1.095 ***	1.266 ***	0.772 ***	0.964 ***	0.712 ***	0.752
Intermediate	0.547 ***	0.741 ***	0.711 ***	0.722 ***	0.544 ***	0.498 ***	0.366 **	0.302
High Education ^a^	--	--	--	--	--	--	--	--
Employed ^a^	--	--	--	--	--	--	--	--
Unemployed	0.992	1.112	0.563 ***	0.547 ***	0.675 ***	0.561 ***	0.249 *	0.173
Student	0.471	2.419 **	0.695	0.289	−0.782	0.159	−0.349 **	−0.061
Pensioner	0.886 ***	0.803 ***	1.473 ***	1.145 ***	3.754 ***	3.008 ***	4.239 ***	3.965
Homemaker	0.954	0.610 ***	0.759 *	0.337 ***	0.949 **	0.181 **	0.610	0.072
Other	0.302	0.277	1.166 ***	0.836 ***	1.613 ***	0.151	0.877 **	−0.245
Daily smoker	−0.172 **	−0.034	−0.032	0.011	0.109 *	0.313 ***	0.190 *	0.500
Overweight	−0.068	0.278 ***	−0.042	0.260 ***	0.026	0.359 ***	0.043	0.216
Sedentarism	0.644 ***	0.398 ***	0.377 ***	0.137 ***	0.344 ***	0.063	0.639 ***	0.111
Intercept	−22.772 ***	−23.498 ***	−18.393 ***	−11.473 ***	4.038	−9.741 ***	−1.883	−5.574
**Random Effects**								
σ	0.570	0.443	0.486	0.554	0.570	0.494	0.606	0.419
Prob > chibar2	0.00	0.00	0.00	0.00	0.00	0.00	0.00	0.00

***, **, and * denote that the explanatory variables are statistically significant at the 99%, 95%, and 90% levels. ^a^ Reference variable.

**Table 4 healthcare-10-00365-t004:** Low self-rated health by sub-samples of countries according to the distribution of Gini tertiles: Oaxaca.

	Silent Generation			Baby Boomers			X Generation			Y Generation		
	Gini T1	Gini T2	Gini T3	Gini T1	Gini T2	Gini T3	Gini T1	Gini T2	Gini T3	Gini T1	Gini T2	Gini T3
**Men**	0.669 ***	0.671 ***	0.750 ***	0.472 ***	0.350 ***	0.480 ***	0.184 ***	0.085 ***	0.179 ***	0.091 ***	0.026 ***	0.084 ***
**Women**	0.792 ***	0.780 ***	0.836 ***	0.500 ***	0.403 ***	0.559 ***	0.217 ***	0.149 ***	0.196 ***	0.104 ***	0.080 ***	0.094 ***
**Difference**	−0.122 ***	−0.109 ***	−0.087 ***	−0.028	−0.054 ***	−0.080 ***	−0.034 **	−0.065 ***	−0.017 ***	−0.014	−0.054 ***	−0.010 **
*Endowment*	−0.057 ***	−0.053 ***	−0.018 ***	−0.042 ***	−0.015	−0.020 ***	0.034 ***	−0.041 ***	0.033 ***	0.012	−0.008	0.013 ***
*Coefficients*	−0.066 ***	−0.069 ***	−0.068 ***	0.009	0.020	−0.027 **	−0.009	−0.044 *	−0.002	−0.017	−0.057 ***	−0.004
*Interaction*	0.000	0.014	−0.001	0.006	−0.059 ***	−0.034 ***	−0.059 ***	0.020	−0.047 ***	−0.009	0.012	−0.019 *

***, **, and * denote that the explanatory variables are statistically significant at the 99%, 95%, and 90% levels.

## Data Availability

Database by waves is available on Internet: European Health Interview Survey Web Page: https://ec.europa.eu/eurostat/web/microdata/european-health-interview-survey (Accesed date: 18 January 2022).

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
