# Peer review of "Gender Gap in Self-Rated Health: A Cohort Perspective in Eastern European Countries"

_healthcare, 2022, doi:10.3390/healthcare10020365_

Round 1

Author Response

We would like to thank Reviewer 1 for the comments and suggestions we have received. We have taken them into account one by one, and we are happy to share with you that not only the quality of the paper has improved considerably from the first version, but we have also enjoyed and learned a lot from this review process.

To make the piece more ground breaking than it currently is, the authors should consider focusing upon some of the inconsistences in the data that they collected (e.g. They mention…and data shows…that males have worse perceptions of their health in some generation cohorts, but women have worse perceptions of their health in other generation cohorts. But, they do not provide any explanations or presumptions about WHY this might be the case.)

The gender gap in self-assessed health and its evolution through generations is a complex issue to explain. It is difficult to give the specific causes of the phenomena for several reasons. In the first place, it is necessary to control the information that men and women take into account in relation to objective and subjective health indicators (Gómez Costilla, García-Prieto, & Somarriba-Arechavala, 2021). This could be different (Benyamini, Leventhal, and Leventhal 2000), or the same (Zajacova, Huzurbazar, and Todd 2017). To this debate we must add the contextual importance, because when the geographical area is considered, the social and cultural differences between countries can bias the way in which individuals perceive their health (Jürges 2007). In our article, we analyze this issue integrating gender differences and the background of Eastern countries. We have also added some new paragraphs to the main text that are intended to be more informative in this explanation.

References:

  • Benyamini, Y., Leventhal, E. A., & Leventhal, H. Gender differences in processing information for making self-assessments of health. Psychosomatic Medicine. 2000; 62(3): 354-64.
  • Gómez-Costilla, P., García-Prieto, C. & Somarriba-Arechavala, N. Aging and Gender Health Gap: A Multilevel Analysis for 17 European Countries. Social Indicators Research. 2021. https://doi.org/10.1007/s11205-020-02595-2
  • Jürges, H. True health versus response styles: exploring cross-country differences in self-reported health. Health Economics. 2007; 16(2): 163-78.
  • Zajacova, A., Huzurbazar, S., & Todd, M. Gender and the structure of self-rated health across the adult life span. Social Science and Medicine. 2017; 187: 58-66.

Throughout the article we have included the following paragraphs:

Introduction

Eastern and Southern countries share a deeper gender gap than the rest of the EU countries [28]. Sociological, political, cultural elements in common due to a similar history during the 20th century could be the roots of a traditional and conservative role of women that identifies them as the person who should be in charge of caring for the family.

Discussion

In addition to work activity (first shift) and housework (second shift), women have to deal with other issues, such as solidarity, calendar management, diversity and equity work, family and friends wellbeing (thirds shift) [49]. Each of these changes can generate gender difficulties, but together they contribute to generating stress and discomfort [50, 51]. Its content and duties can vary across generations and cultures.

However, in these Eastern economies, health care resources are limited and the universalism of access, typical of communism, is lost [52]. The transition to market economies has evolved in parallel with the deterioration of public health resources [53]. The adult population reports poorer self-rated health compared to other European countries [54]. The fact that the gender gap is higher in the elderly population of the Eastern countries [55] and lower in other Bismarkian countries, could be explained by the protective effect of the pension system, the provisions of social resources and the planning ahead [56].

1. At the very start of the article, the authors suggest that low self-assessments of one’s health leads “to worse indicators of acute and chronic morbidity, a greater degree of disability, a more frequent use of health resources, deficiencies and/or difficulties in the accomplishment of the daily work, domestic and 40 recreational activities.” It is important, here, to remove any implications that these conditions and behaviors are consequences of the low self-assessments. Research methodologists stress the importance of considering causal-time order when making associations between variables. Although it may be true that the assessment of one’s own health actually impacts his or her health and daily functioning, the opposite could also be true. Those health issues in the first place may recognize that they have poor health, thus giving themselves low assessments. The point is that we can’t be sure which of the variables causes the other.

We agree with the casual relationship that could be inferred about these interrelation variables. To avoid this problem, we have deleted the last part of this paragraph: “as it is widely regarded as a good predictor of both health care use and mortality”.

2. In line 134, the authors refer to controlling for similarities. I believe, however, that they mean to state that they are controlling for differences. Unless I am mistaken, the focus on the specific countries identified is intended to eliminate confounding variables (differences) that may explain variations in data.

To improve the explanation of the following paragraph: “We have focused on Eastern European countries to control for similarities (welfare state and capacity to create richness), in order to understand how differences (equity) lead to different health status”, we have rewritten it as: “The selection of countries that share important characteristics (welfare state and capacity to create richness), allows us to identify how differences (equity) lead to different health status”.

3. Graph 2, in my opinion, makes the most interesting points in the article. Elaboration on this graph, however, does not appear until the authors present Table 4. It seems that, for readers to fully understand distinctions in the data presented in Graph 2

The tables and graphs included in the Descriptive results section provide us with an overview of the research questions. Tables included in the Empirical strategy section provide us with further information about causal effects and their statistically significance. Therefore, Table 4 ratifies information provided in Graph 2, but it also allows us to validate the starting theoretical hypotheses.

4. The information presented in lines 217-241 describe the methods used to analyze data, not results. Therefore, it should appear in the “Materials and Methods” portion of the article.

As reviewer suggests we have moved these paragraphs to Materials and Methods section.

5. In lines 312-213, the authors refer to age differences within generations. The article, however, presents no comparison of subjects within each generation based upon their specific ages. How, then, is this data obtained?

The main objective of this article is to analyze the gender gap in self-reported health from a cohort perspective. Age is included as explanatory variables in Table 3, in the same way as other variables such as marital status or educational level. Table 4 allows us to identify whether gender differences in self-perceived health is due to the endowment and/or coefficient effect of the explanatory variables introduced in Table 3 (including age). In the same way as in Table 3, the estimates have been repeated for sub-samples of generational cohorts.

PRESENTATION

Thanks very much for this editing work. We have taken them into account one by one

  1. The phrase “and with” can be removed from line 65 so that the sentence reads, “In the European context, data from the European Social Survey…”
  2. The word “to” can be removed from line 85 so that the sentence reads, “Education attainments benefit more women than men …”
  3. The phrase, “related to” should be added to line 152 so that the sentence reads, “We select primary questions related to whether people report
  4. The word, “about” should be added to line 162 so that the sentence reads, “Nevertheless, their inclusion provides further empirical evidence about whether…”
  5. The word “other” in line 177 should be changed to “another.”
  6. The word “educative” in lines 179 and 181 should be changed to “education.”
  7. The word “smoothed,” in line 190 doesn’t really make sense in the context of the sentence. Can the authors use a different word?
  8. The sentence in line 191 should be rephrased to state, “As expected, women consider their state of health as worse than men…”
  9. The word “worsen” in line 192 should be changed to “worse.”
  10. The words, “A,” and “the” should be added in line 277 so that the sentence reads, “A special case are male smokers from the silent generation
  11. The word, “the” should be added to in line 305 so that the sentence reads, “…along the generations…”
  12. The word “smoother,” in line 305 doesn’t really make sense in the context of the sentence. Can the authors use a different word?
  13. The sentence in line 306 should be rephrases to state, “The coefficient effect indicates a negative role increasing gender…”
  14. The phrase “worse value” should be changed to “poorly assess,” so that the sentence reads, “…people who poorly assess their health…”
  15. The word “the” should be added to line 341 so that it reads, “However, the silent generation is also…”
  16. The word, “authors,” in line 380 should be changed to “potential researchers,” so that the sentence reads, “Potential researchers should discuss the results…”

Reviewer 2 Report

This paper addresses a significant issue: the gender gap in self-rated health in the region of Eastern Europe. It contributes and adds to the existing literature on Psychology, Sociology, Medicine, and Public Health, as well as broadens the discussion about gender equality in the health system. The paper is also well-structured.

Nevertheless, I would suggest some changes to improve the research.

Firstly, I would link the research with the so-called third shift of women (first shift at work, second shift at home, and third shift in mental terms –that work behind the scenes that make anyone in the family show up to anything on time). Different literature considers that the third shift explains the gender gap in health.

Secondly, I would explain that Eastern European countries are a group of countries with sociological, political, and cultural elements in common due to a similar history during the XX century, which is at the roots of this conception that women must be in charge of family care. 

I would also pay some attention to the English grammar. 

Author Response

This paper addresses a significant issue: the gender gap in self-rated health in the region of Eastern Europe. It contributes and adds to the existing literature on Psychology, Sociology, Medicine, and Public Health, as well as broadens the discussion about gender equality in the health system. The paper is also well-structured.

We are very grateful for the positive feedback from Reviewer 2, because it gives us the strength to improve our work. As social researchers, our desire is to analyse health from a multidisciplanary perspective and to provide scientific information to reduce the gender gap.

Nevertheless, I would suggest some changes to improve the research.

Firstly, I would link the research with the so-called third shift of women (first shift at work, second shift at home, and third shift in mental terms –that work behind the scenes that make anyone in the family show up to anything on time). Different literature considers that the third shift explains the gender gap in health.

We appreciate the suggestion; thus we have added the following paragraph in Discussion section:

In addition to work activity (first shift) and housework (second shift), women have to deal with other issues, such as solidarity, calendar management, diversity and equity work, family and friends wellbeing (thirds shift) [49]. Each of these changes can generate gender difficulties, but together they contribute to generating stress and discomfort [50, 51]. Its content and duties can vary across generations and cultures.

References:

  • Oppenheimer L. The Surprise Origin of Women's Mental Load. Available at: https://www.brighthorizons.com/employer-resources/mental-load-starts-at-the-office Accessed January 30, 2022
  • Gerstel, N. The Third Shift: Gender and Care Work, Outside Home Qual. Soc., 2020; 23: 4.
  • Santhost, L., Keetnan, B.P, Jain, S. The “Third Shift”: A Path Forward to Recognizing and Funding Gender Equity Efforts. J. Womens Health (Larchmt 2020; 29(11): 1359–60.

Secondly, I would explain that Eastern European countries are a group of countries with sociological, political, and cultural elements in common due to a similar history during the XX century, which is at the roots of this conception that women must be in charge of family care.

We like very much your recommendation, so we have included this new paragraph in the Introduction section:

Eastern and Southern countries share a deeper gender gap than the rest of the EU countries [28]. Sociological, political, cultural elements in common due to a similar history during the 20th century could be the roots of a traditional and conservative role of women that identifies them as the person who should be in charge of caring for the family.

I would also pay some attention to the English grammar

We have requested a professional English proofreading

Reviewer 3 Report

  1. The paper should be professionally edited for English. While the meaning is generally clear, the choice of words is often less than ideal.
  2. There may be issues with common method variance that a post-hoc test may be able to elucidate. I suggest a Harman single factor test with appropriate discussion at the end of the methods/data presentation
  3. Further discussion of the contextual issues associated with Eastern European health issues in general, and any relevant country-specific health issues within your sample, can be added as an additional section in the literature discussion.
  4. The data is quite old - are there any recent datasets that can supplement these findings and confirm if the trends have been changed or are confirmed over time?

Author Response

1. The paper should be professionally edited for English. While the meaning is generally clear, the choice of words is often less than ideal.

Thanks for the suggestion. We have requested a professional English proofreading. We hope that the changes reflected in this new version are sufficient to improve the article to your satisfaction.

2. There may be issues with common method variance that a post-hoc test may be able to elucidate. I suggest a Harman single factor test with appropriate discussion at the end of the methods/data presentation

We take note of the Harman single factor test, which we did not know. This evaluation process has allowed us to learn and update our knowledge of applied techniques. So far we understand the importance of Harman single factor test is to identify method bias when the same measurement instrument is used to collect data for both dependent and independent variables. In our case we have combined micro and macro data from different databases. Besides, we have just focused on Eastern countries because the selection of countries that share important characteristics (such as welfare state), allows us to identify how differences (such as in equity) lead to different health status. The consideration of citizens from countries with different welfare states would have introduced technical problems about what is considered good health for a Nordic citizen and one from an Eastern country. Therefore, we understand that for our sub-sample it is not necessary to identify the method bias, but we are very grateful for the comment that has made us reflect on its convenience.

3.Further discussion of the contextual issues associated with Eastern European health issues in general, and any relevant country-specific health issues within your sample, can be added as an additional section in the literature discussion.

We appreciate your suggestion; thus we have added the following paragraphs to highlight the contextual influence:

Introduction

Eastern and Southern countries share a deeper gender gap than the rest of the EU countries [28]. Sociological, political, cultural elements in common due to a similar history during the 20th century could be the roots of a traditional and conservative role of women that identifies them as the person who should be in charge of caring for the family. Its content and duties can vary across generations and cultures.

Discussion

However, in these Eastern economies, health care resources are limited and the universalism of access, typical of communism, is lost [52]. The transition to market economies has evolved in parallel with the deterioration of public health resources [53]. The adult population reports poorer self-rated health compared to other European countries [54]. The fact that the gender gap is higher in the elderly population of the Eastern countries [55] and lower in other Bismarkian countries, could be explained by the protective effect of the pension system, the provisions of social resources and the planning ahead. long term [56].

  • Eikemo, T. A., Bambra, C., Judge, K., & Ringdal, K. Welfare state regimes and differences in self-perceived health in Europe: A multilevel analysis. Soc Sci Med. 2008; 66(11): 2281-2295.
  • Deacon, B. Deacon, &B. (2000). Eastern European welfare states: The impact of the politics of globalization. Journal of European Social Policy. 2000; 10(2): 146-61.
  • Guarnizo-Herreño, C., Watt,R.G., Stafford, M., Sheiham, A., & Tsakos, G. Do welfare regimes matter for oral health? A multilevel analysis of European countries, Health & Place. 2017; 46: 65-72
  • Obrizan, M. (2018). Quantifying the gap in self-rated health for transition countries over 1989–2014. Comparative Economic Studies. 2018; 60(3): 388-409.
  • Gómez-Costilla, P., García-Prieto, C. & Somarriba-Arechavala, N. Aging and Gender Health Gap: A Multilevel Analysis for 17 European Countries. Social Indicators Research. 2021. https://doi.org/10.1007/s11205-020-02595-2

4. The data is quite old - are there any recent datasets that can supplement these findings and confirm if the trends have been changed or are confirmed over time?

The European Health Interview Survey (EHIS) is conducted every 5 five year. The second wave (EHIS 2) took place between 2013 and 2015 in all EU Member States, Iceland and Norway. In 2014 this survey was implemented in: Bulgaria, Czechia, Estonia, Greece, Spain, France, Croatia, Italy, Cyprus, Latvia, Lithuania, Luxembourg, Hungary, Malta, Netherlands, Austria, Poland, Portugal, Romania, Slovenia, Slovakia, Finland and Sweden.

To date there are no more recent data available with this instrument: https://ec.europa.eu/eurostat/web/microdata/european-health-interview-survey

However, the data is interesting on its own because it was compiled before the economic crisis and pandemic situation. In any case, as soon as new data is available, we would like to update this information.